# Combined General/Epidural Anesthesia vs. General Anesthesia on Postoperative Cytokines: A Review and Meta-Analysis

**DOI:** 10.3390/cancers17101667

**Published:** 2025-05-15

**Authors:** Erica J. Lin, Stephen Prost, Hannah J. Lin, Syed Shah, Ru Li

**Affiliations:** Department of Anesthesiology, Health Science Center, Stony Brook University, Stony Brook, NY 11794-8480, USA; linerica6213@gmail.com (E.J.L.); stephen.probst@stonybrookmedicine.edu (S.P.); hannahlin545@gmail.com (H.J.L.); syed.shah@stonybrookmedicine.edu (S.S.)

**Keywords:** epidural analgesia, inflammation, immune response, stress response, regional anesthesia

## Abstract

The present study aimed to analyze and compare potential associations between the usage of general or combined general/epidural anesthesia and postoperative cytokines. Given the association between inflammatory response and various complications after surgery, we narrowed our focus to pro-inflammatory and regulatory cytokines. We performed a meta-analysis on 21 studies, finding that serum levels of IL-6, TNF-α, CRP, and other pro-inflammatory cytokines were significantly less for combined epidural and general anesthesia compared to general anesthesia alone. Our results provide an alternative explanation for the benefit of epidural analgesia on postoperative outcomes. Further studies on the effect of epidural anesthesia on postoperative cytokines are required to assess this relationship reliably, as well as the potential application to clinical settings.

## 1. Introduction

Surgical interventions evoke a stress response in patients due to tissue trauma, which is characterized by endocrine alterations, including an increase in plasma cortisol and activation of the sympathetic nervous system with secretion of endogenous catecholamines [1]. The local tissue injury during surgical and perioperative events such as blood transfusion also initiates a systemic inflammatory response [2]. The surge of pro-inflammatory cytokine release, detectable in peripheral blood during the immediate postoperative period has been implicated in the development of various complications, including myocardial injury, acute kidney injury and postoperative delirium [3,4]. The change in postoperative cytokine profiles may also increase the risk of infectious complications [5,6]. Studies have also shown that postoperative inflammation is a risk factor for postoperative mortality, independent from the occurrence of postoperative complications [7,8]. Current research suggests that the choice of anesthetics and postoperative analgesia may alter the profile of the postoperative immune response [9,10] and could affect long term outcomes after surgery [11,12,13].

Epidural anesthesia or analgesia (EA) is a well-established regional anesthesia technique for postoperative analgesia and may be used for surgical anesthesia in selected procedures. Administration of local anesthetic in the epidural space prevents neurotransmission through the spinal nerves, thus preventing transmission pain, as well as other sensory, motor, and sympathetic impulses [14]. It has been suggested that EA may reduce cortisol production and maintain a physiological cytokine balance during postoperative period [15]. Additionally, the use of EA could decrease the demand for systemic opioids, thereby reducing the negative effect of opioids on cell-mediated immunity [16]. Despite several clinical trials investigating the effect of epidural analgesia on perioperative inflammatory mediators, no meta-analysis has been performed on the topic. Evaluation of these changes across various anesthesia methods will help to optimize recovery and guide postoperative care. Therefore, we conducted a meta-analysis on the postoperative inflammatory responses after general anesthesia (GA) compared to GA plus EA.

## 2. Materials and Methods

### 2.1. Study Objectives

The objective of the study is to compare the postoperative serum cytokine levels in patients who had general anesthesia plus EA to those who had general anesthesia only with postoperative systemic analgesia. Our primary outcomes of interest are serum levels of interleukin-6 (IL-6), C-reactive protein (CRP), and tumor necrosis factor-α (TNF-α) 24 h after surgery. Our secondary outcomes of interest include CRP, TNF-α, IL-1β, IL-4, IL-8, IL-10, and cortisol levels immediately after surgery and 24 h postoperatively.

### 2.2. Search Strategy and Study Selection Criteria

We followed the Preferred Reporting Items for Systematic reviews and Meta-analysis (PRISMA) statement for conducting and reporting the meta-analysis [17]. The study was registered in the “International Prospective Register of Systematic Reviews” (PROSPERO) in 2019 (CRD42019145596). We systematically searched the PubMed, Central, EMBASE, CINAHL, Google Scholar, and Web of Science citation indexes for relevant studies. Used terms included ‘epidural’, ‘interleukin’, and ‘cytokine’, and their Boolean combinations, the full search strategy is included in the Appendix A. We did not impose any language restriction at the time of the literature search. Two authors independently conducted all the searches and discrepancies were discussed after the search process. The last search was carried out on 2 January 2025.

Studies were included if they fulfill the following criteria:

Study design and patients: randomized control trials in adult patients undergoing cancer or non-cancer surgery under general anesthesia.

Intervention: epidural catheter for perioperative analgesia.

Control: general anesthesia only without epidural analgesia.

Outcome: studies need to report at least one of the primary outcomes to be included.

Comparison must be reported as mean ± standard deviation. Alternatively, it must be possible to derive mean ± standard deviation from the reported data. Studies which did not include data in a suitable format were excluded from the analysis. Exclusion criteria were studies with children under 18, ongoing clinical trials, study designs other than RCTs, studies on subjects who did not have surgery under general anesthesia, and studies where the patient received regional anesthesia other than EA.

### 2.3. Data Extraction

The following information was extracted from each study: bibliographical information (family name of the first author, year of publication, and PubMed ID), study design (number of participants in the GA + EA and EA cohort and time points for outcome measurement), and outcomes (plasma level of cytokine, interleukin, and stress hormone). When study results are only displayed as graphical form, two authors independently extracted the data using WebPlot Digitizer [18]. For the studies where the plasma cytokine values were expressed as “medium” and no standard deviation values were reported, the mean ± standard deviation values were obtained using ‘Deep Meta tool Version 1.0’ [19]. Data extraction was conducted using standardized pro-forma and checked by a second author (RL and EL).

We used the Cochrane Collaboration tool 2.0 for assessing the risk of bias in each included study [20]. Two authors performed the assessment independently at the same time and any disagreements were discussed with and resolved by a third author. Studies were assessed on randomization, deviation from intended intervention, missing outcome data, outcome measurement, and selective reporting. Each category of the study was assigned ‘low risk’, ‘high risk’, or ‘some concerns’.

### 2.4. Statistical Analyses

We conducted meta-analysis for outcomes reported in more than five studies. Review Manager V5.3. (Cochrane Collaboration, Copenhagen, Denmark) was used for the pooled analysis. As the effect size for the outcomes are of clinical relevance, for continuous variables, we calculated the standardized mean differences (SMD) based on the random effect model. The random effect model was used in the analysis due to varying effect sizes between studies and the inherent heterogeneous nature of the surgical procedure and patient population involved. In addition, we conducted subgroup analyses of the primary outcomes, including only cancer surgeries. For outcomes that reported positive findings, Duval and Tweedie’s trim and fill was used to assessed the publication bias and Egger’s regression was used to assess for small study effect; both were conducted using methods described by Suurmond et al. [21]. For all outcomes, the statistical significance was set to *p* < 0.05 and with 95% confidence intervals. We used GRADEpro Guideline Development Tool (GRADEpro GDT, McMaster University, 2015) to assess the quality of the meta-analysis findings.

## 3. Results

### 3.1. Description of Included Studies

Following the search criteria, a total of 355 studies were shortlisted, of which 34 passed the title and abstract screening (Figure 1). Fourteen were removed on further reading since four studies did not report the primary outcome we selected, two studies did not report the details of epidural procedure, one study reported LPS-induced TNF-α rather than serum level, one was a retrospective study, one study designed to compare different local anesthetics in epidural procedure, and one study did not report cytokine levels at the time point we chose. Finally, we included 21 studies in our meta-analysis and their characteristics are described in Table 1 [22,23,24,25,26,27,28,29,30,31,32,33,34,35,36,37,38,39,40,41,42]. Among them, 15 are cancer surgeries and 7 are non-cancer surgeries. The risk of bias assessment for each study is shown in Figure 2. The most common source of bias came from the randomization process since most studies did not report information about how the allocation sequence was concealed.

### 3.2. IL-6 24 Hours After Surgery

There were 16 studies which reported plasma levels of IL-6 24 after surgery [22,23,24,25,27,28,29,30,32,35,36,37,38,39,40,41]. This included a total of 480 patients who received epidural analgesia and 485 patients who only received general anesthesia. The pooled results showed statistically lower IL-6 levels in the EA + GA cohort in comparison to the GA cohort [standardized mean difference (SMD) = −1.99, 95% confidence interval (CI) = −2.67 to −1.31, I^2^ = 97%, Egger’s regression *p* < 0.01, trim and fill reported five missing studies, Figure 3, Appendix A]. Sensitivity analysis did not alter the finding when each of the studies was removed. Quality of evidence is very low due to significant data heterogeneity, as well as high risk of publication bias.

To address the heterogeneity, we conducted subgroup analyses based on surgery type and based on the length of surgery. We found that thoracic and upper gastrointestinal surgery groups were not significantly different from abdominal/pelvic procedures, and surgeries longer than 180 min were comparable to shorter surgeries. Neither subgroup analyses significantly reduced the heterogeneity (Appendix A).

We also conducted a subgroup analysis, including only the studies involving cancer surgery. This included twelve studies [23,24,27,28,30,32,35,37,38,39,40,41], and the IL-6 level was significantly lower in EA + GA cohort (SMD = −1.91, 95% CI = −2.60 to −1.21, I^2^ = 93%, Egger’s regression *p* < 0.01, trim and fill reported three missing studies, Appendix A). Sensitivity analysis did not alter the finding when each of the studies was removed. Quality of evidence is very low due to significant data heterogeneity, as well as high risk of publication bias.

### 3.3. TNF-α 24 Hours After Surgery

There were nine studies which reported serum TNF-α levels 24 h after surgery, this included a total of 355 patients who received epidural analgesia and 357 patients who only received general anesthesia [23,24,29,30,31,32,33,36,41]. The pooled results showed significantly lower TNF-α levels in the EA cohort (SMD = −1.13, 95% CI = −1.74 to −0.51, I^2^ = 93%, Egger’s regression *p* = 0.11, trim and fill reported one missing study, Figure 4). Sensitivity analysis did not alter the finding when each of the studies was removed. Quality of evidence is low due to significant data heterogeneity.

The cancer surgery only subgroup analysis included seven studies [23,24,30,31,32,33,41], and the TNF-α levels was significantly lower in EA cohort (SMD = −0.89, 95% CI = −1.47 to −0.32, I^2^ = 89%, Egger’s regression *p* = 0.05, trim and fill reported no missing studies, Appendix A). Sensitivity analysis did not alter the finding when each of the studies was removed. Quality of evidence is low due to significant data heterogeneity.

### 3.4. CRP 24 Hours After Surgery

There were seven studies which reported serum CRP levels 24 h after surgery, this included a total of 239 patients who received epidural analgesia and 242 patients who only received general anesthesia [24,26,31,33,34,40,41]. The pooled results showed significantly lower CRP levels in the EA cohort (SMD = −0.61, 95% CI = −0.97 to −0.24, I^2^ = 71%, Egger’s regression *p* = 0.11, trim and fill reported no missing studies, Figure 5). Sensitivity analysis did not alter the finding when each of the studies was removed. Quality of evidence is low due to significant data heterogeneity.

The cancer surgery only subgroup analysis included six studies [24,26,31,33,40,41], and the CRP level was significantly lower in EA cohort (SMD = −0.72, 95% CI = −1.07 to −0.36, I^2^ = 63%, Egger’s regression *p* = 0.06, trim and fill reported no missing studies, Appendix A). Sensitivity analysis did not alter the finding when each of the studies was removed. Quality of evidence is medium due to data heterogeneity.

### 3.5. IL-6 at the End of Surgery

There were 11 studies which reported plasma levels of IL-6 immediately after surgery [22,23,24,27,28,29,35,36,39,41,42]. The pooled results showed statistically lower IL-6 levels in the EA + GA cohort in comparison to the GA cohort (SMD = −1.26, 95% CI = −2.22 to −0.30, I^2^ = 96%, Egger’s regression *p* < 0.01, trim and fill reported no missing studies, Appendix A). Sensitivity analysis reported non-significant pooled SMD when the study by Fares et al. was removed [27]. Quality of evidence is very low due to significant data heterogeneity and sensitivity analysis finding.

The cancer surgery only subgroup analysis included seven studies, and the IL-6 level was significantly lower in EA cohort (SMD = −1.75, 95% CI = −2.61 to −0.9, I^2^ = 93%, Egger’s regression *p* = 0.01, trim and fill reported no missing studies, Appendix A) [23,24,27,28,35,39,41,42]. Sensitivity analysis did not alter the finding when each of the studies was removed. Quality of evidence is low due to significant data heterogeneity.

### 3.6. TNF-α at the End of Surgery

There were seven studies which reported serum TNF-α levels immediately after surgery [23,24,29,31,33,36,41]. The pooled results showed significantly lower TNF-α levels in the EA cohort (SMD = −0.80, 95% CI = −1.53 to −0.08, I^2^ = 93%, Egger’s regression *p* < 0.01, trim and fill predicted no missing studies, Appendix A). The difference appears to be driven primarily by one study by Hadimioglu et al. [29], and exclusion of the said study in the sensitivity analysis resulted in non-significant pooled SMD. Quality of evidence is very low due to significant data heterogeneity and sensitivity analysis finding.

Five of the included studies were conducted on cancer patients, the pooled data did not demonstrate significant difference between EA and GA cohorts (SMD = −0.08, 95% CI = −0.39 to 0.23, I^2^ = 54%, Appendix A) [23,24,31,33,41].

### 3.7. CRP at the End of Surgery

There were five studies which reported serum CRP levels at the end of surgery [24,31,33,34,41]. The pooled results showed significantly lower CRP levels in the EA cohort (SMD = −1.60, 95% CI = −2.78 to −0.42, I^2^ = 96%, Egger’s regression *p* = 0.15, trim and fill predicted no missing studies, Appendix A). The difference appears to be driven primarily by one study by Duque et al. [24], and exclusion of the said study in the sensitivity analysis resulted in non-significant findings. Quality of evidence is low due to significant data heterogeneity.

Four of them were conducted on cancer surgery [24,31,33,41], and the lower CRP levels in the EA cohort again appear to be driven primarily by the study Duque et al. [24]. (Appendix A).

### 3.8. Cortisol Levels After Surgery

There were eight studies which reported serum cortisol levels 24 h after surgery [23,24,28,30,34,35,39,41], this included a total of 270 patients in each cohort. The pooled results showed significantly lower cortisol levels in the EA cohort (SMD = −0.82, 95% CI = −1.22 to −0.43, I^2^ = 78%, Egger’s regression *p* = 0.86, trim and fill predicted no missing studies, Appendix A). Sensitivity analysis did not alter the findings when each of the studies was removed. Quality of evidence is low due to significant data heterogeneity.

There were seven studies which reported serum cortisol levels at the end of surgery [23,24,28,34,35,39,41]. The pooled results showed significantly lower cortisol levels in the EA cohort (SMD = −1.94, 95% CI = −3.19 to −0.68, I^2^ = 96%, Egger’s regression *p* < 0.01, trim and fill predicted no missing studies, Appendix A). Sensitivity analysis did not alter the finding when each of the studies was removed. Quality of evidence is low due to significant data heterogeneity.

### 3.9. Other Interleukin Levels After Surgery

There were five studies which reported serum IL-1β levels 24 h after surgery, this included a total of 187 patients who received epidural analgesia and 179 patients who only received general anesthesia [24,31,32,33,41]. The pooled results showed significantly lower IL-1β levels in the EA cohort (SMD = −1.70, 95% CI = −2.66 to −0.74, I^2^ = 93%, Egger’s regression *p* = 0.47, trim and fill predicted no missing studies, Appendix A). Sensitivity analysis did not alter the finding when each of the studies was removed. Quality of evidence is medium due to data heterogeneity.

There were five studies which reported serum IL-4 levels 24 h after surgery, this included a total of 112 patients who received epidural analgesia and 111 patients who only received general anesthesia [23,24,28,32,37]. The pooled results showed no difference between the two cohorts (SMD = 0.83, 95% CI = −1.26 to 2.91, I^2^ = 93%, Egger’s regression *p* = 0.13, trim and fill predicted no missing studies, Appendix A). Sensitivity analysis did not alter the finding when each of the studies was removed. Quality of evidence is low due to significant data heterogeneity.

There were six studies which reported serum IL-8 levels 24 h after surgery, this included a total of 219 patients who received epidural analgesia and 220 patients who only received general anesthesia [22,24,25,27,31,33]. The pooled results showed significantly lower IL-8 levels in the EA cohort (SMD = −1.53, 95% CI = −2.04 to −1.03, I^2^ = 79%, Egger’s regression *p* = 0.84, trim and fill predicted no missing studies, Appendix A). Sensitivity analysis did not alter the finding when each of the studies was removed. Quality of evidence is low due to significant data heterogeneity.

There were five studies which reported serum IL-10 levels 24 h after surgery, this included a total of 149 patients who received epidural analgesia and 148 patients who only received general anesthesia [30,32,35,38,41]. The pooled results showed no difference between the two cohorts (SMD = 1.06, 95% CI = −0.52 to 2.64, I^2^ = 79%, Egger’s regression *p* < 0.01, trim and fill predicted no missing studies, Appendix A). Sensitivity analysis did not alter the finding when each of the studies was removed. Quality of evidence is low due to significant data heterogeneity.

## 4. Discussion

Epidural anesthesia is one of the first regional anesthesia techniques conceived and was historically the ‘gold standard’ for postoperative analgesia after major truncal surgeries [43]. Despite the development of peripheral nerve blocks in recent years, epidural analgesia remains in widespread use [44]. Epidural infusion of local anesthetics offers excellent relief for postoperative pain, reduces postoperative pain and the need for systemic opioid analgesia, which is associated with a myriad of complications, including immune response alteration [44]. In addition to its effect on the somatic sensory (and to a lesser extent, motor nerves), the spread of local anesthetics also affects the sympathetic trunk near the infusion site [14]. Sympathetic blockade is a known effect of epidurals and can be associated with intraoperative hypotension. However, they also blunt the extent of sympathetic response to surgery, and have been shown to reduce postoperative cardiac events [45]. As the immune response is intricately linked to the neuroendocrine system, it is physiologically feasible that blunted perioperative stress response through the use of epidural could alter postoperative immune response.

In this meta-analysis, we found that EA + GA was associated with lower levels of IL-6, TNF-α, and CRP, as well as other pro-inflammatory cytokines, postoperatively. This supports our hypothesis that in addition to alleviating postoperative pain and sympathetic activation, the use of EA may also reduce the pro-inflammatory response after surgery. Tissue damage from surgical manipulation results in local inflammation, which then induces systemic acute-phase response. IL-6 and TNF-α are two cytokines that initiate the early inflammatory responding to tissue injury. TNF-α has a short half-life (4.6 min) and the peak concentration is usually seen within two hours after trauma [5,46]. TNF-α also activates macrophages and is potent chemokine for neutrophil. In addition, the release of TNF-α also stimulates the production of IL-6, which subsequently enters the systemic circulation [47]. As a pleiotropic cytokine, IL-6 (serum half-life of 15.5 h) plays an important role in various physiological and pathological processes, especially in regulating the proliferation and differentiation of T and B lymphocytes and natural killer cells [48,49]. The activity of both TNF-α and IL-6 also stimulates the production of acute phase proteins including CRP and the production of corticotrophin releasing hormones, which upregulate the hypothalamo–pituitary–adrenal axis.

Pro-inflammatory cytokines such as IL-6, TNF-α, and IL-8 are believed to be important mediators in systemic response to surgery and have a high predictive value for the development of postoperative complications such as acute respiratory distress syndrome, systemic inflammatory response syndrome, sepsis, multiple organ failure, and multiple organ dysfunction syndrome. Szczepanik et al. analyzed 99 consecutive patients who underwent gastrectomy, and reported that elevated IL-6 on postoperative day 1 is an independent risk factor for postoperative complications [50]. Similarly, Straatman et al. found that CRP on postoperative day 3 is independently associate with the development of postoperative complications [51].

Cortisol (serum half-life is 60 to 120 min) is a steroid hormone which is elevated by physiological stress [52]. Elevated serum cortisol is commonly observed after surgery, and possible mechanisms include the upregulated synthesis secondary to IL-6 and TNF-α stimulation [53]. Excessive circulating of corticosteroids is associated with impaired wound healing, insulin resistance, as well as immunosuppression, all of which will adversely affect postoperative recovery [54].

IL-4 (19 min after intravenous administration, over 24 h by combing with anti-IL-4 antibody) is a regulatory cytokine that promotes the differentiation of B cells and T cells but downregulates macrophage activity and promotes wound healing [55]. In addition, IL-4 also upregulates the production of IL-10 by T helper cells [56]. IL-10 is also a regulatory cytokine which downregulates macrophage and nuclear factor kappa-light-chain-enhancer of activated B cells (NF-κB) activities. It also reduces the production of TNF-α, Interferon **γ**, and other proinflammatory cytokines [57]. IL-4 and IL-10 are also the main cytokines involved in promoting wound healing [58,59]. While the pooled data suggest that EA cohort had slightly higher serum IL-4 and IL-10 levels, neither were not statistically significant. Considering the small number of studies and the heterogeneous surgical procedures [57], it is not clear if epidural analgesia has a significant effect on the expression of regulatory cytokines.

There is increasing evidence that perioperative events may affect the outcome of cancer surgery. While the exact mechanism is not clear, this may be attributable to the direct effect of surgical manipulation, perioperative medications (including anesthetic agents), postoperative inflammation and immune function changes [60]. It is thought that IL-6/JAK/STAT pathways could affect the proliferation, survival, and metastasis of cancer cells, while blocking the pathway in animal studies reduced the tumor burden [61]. In addition, TNF-α, IL-1β, and IL-6 have been demonstrated to suppress cell mediated immunity and promote intravascular tumor cell adhesion [62,63].

We therefore conducted subgroup analyses, including only the cancer surgery trials. We found that at 24 h after surgery, the EA cohort had significantly lower IL-6, TNF-α, and CRP. The release of proinflammatory cytokines can alter the immune functions and lead to impaired response to malignancy [64]. In cancer surgery, the immunosuppressive effect of general anesthesia and systemic opioids is problematic, considering that the suppressed CMI may affect the recognition of circulating tumor cells and thereby promote the formation of metastatic niche [65]. In such cases, blunted pro-inflammatory response after surgery afforded by EA may subsequently translate to better postoperative outcomes [60].

This meta-analysis has several limitations. There was considerable inter-study heterogeneity across all the outcomes, which reduces the reliability of the evidence. This could be due to the heterogeneity of surgical procedures and postoperative care from the different studies or may represent differences in the assays used in the different studies. Different surgeries involves various levels of surgical trauma and subsequently, the magnitude of cytokine release. For example, the renal transplant surgery (Hadimioglu 2012 [29]) showed significantly higher absolute levels of inflammatory cytokines than other studies since the surgery of renal transplantation is typically a lengthy and more invasive procedure. Immunosuppressants are typically in the real transplant patients to prevent rejection but do not seem to abolish the release of inflammatory cytokines. Another factor to be considered is the usage of opioids. Of the 21 included studies, 4 studies (Elsayed 2020, Fares 2014, Tan 2016, Wang 2019 [25,27,36,37]) combined local anesthetics with opioid during surgery an 5 studies (Gu 2015, Liu 2019, Xu 2014, Xu 2020, Yokoyama 2005 [28,31,38,39,41]) added opioids in the postoperative analgesia. Only one study, Tan 2016 [36], reported a significantly higher increase in IL-6 levels in the GA group compared to the TEA+ GA group, with a difference greater than those observed in other studies. Therefore, opioid use during perioperative and postoperative periods does not appear to be the primary factor contributing to the difference in proinflammatory cytokine levels between TEA + GA and GA-only groups. As most studies only reported cytokines up to 24 h after surgery, there are also insufficient data to determine the duration of effect from epidural analgesia. Another limitation is that while we have demonstrated statistically significant cytokine changes, it is not clear if the immune response changes translate to improved clinical outcomes. While there are clinical evidence linking perioperative inflammatory mediators with postoperative complication, there are limited data on whether altering the inflammatory response leads to better outcomes. Larger studies with longer postoperative follow up periods are required to reliably assess the effect of EA on cytokines on postoperative outcomes. This is especially important for cancer surgery, as recurrences often take years to become apparent.

## 5. Conclusions

In conclusion, this meta-analysis found that epidural analgesia is associated with significantly lower levels of pro-inflammatory cytokines, CRP, and cortisol 24 h after surgery. A similar pattern was also seen in the cancer surgery sub-cohort. The evidence on epidural analgesia and the expression of regulatory cytokines (including IL-4 and IL-10) is insufficient for drawing conclusions. Studies with longer postoperative follow up periods are needed to assess of the change in cytokine profiles are associated with better clinical outcomes.

## Figures and Tables

**Figure 1 cancers-17-01667-f001:**
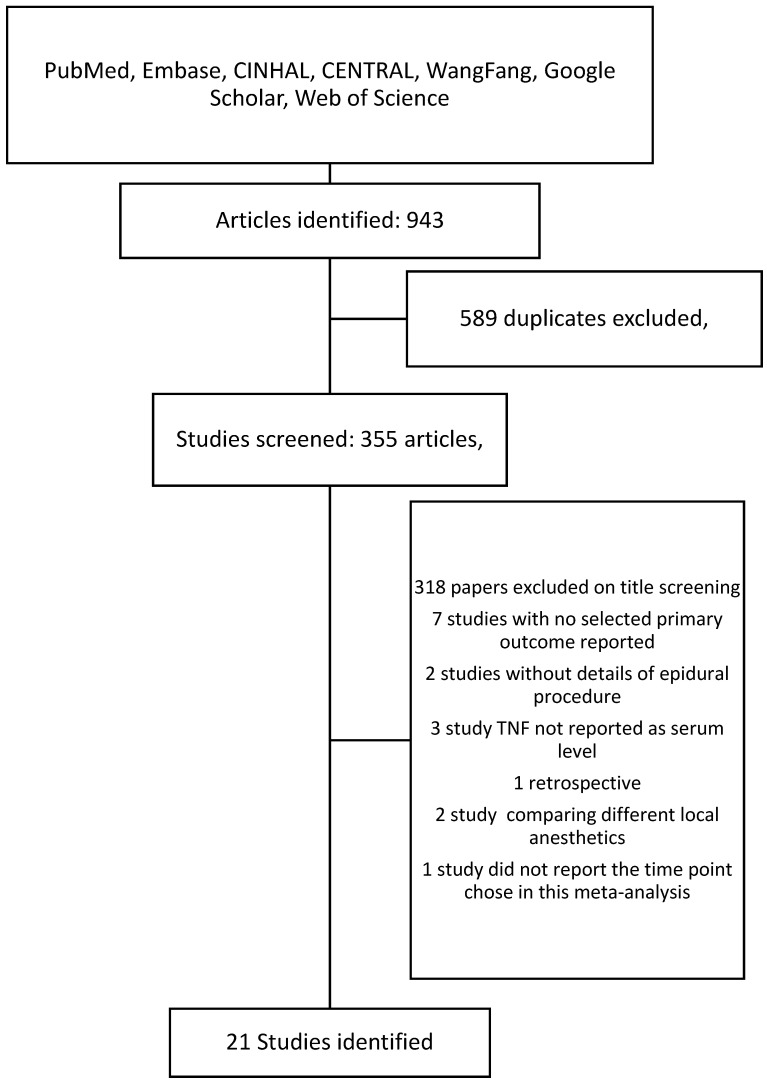
Search flow chart.

**Figure 2 cancers-17-01667-f002:**
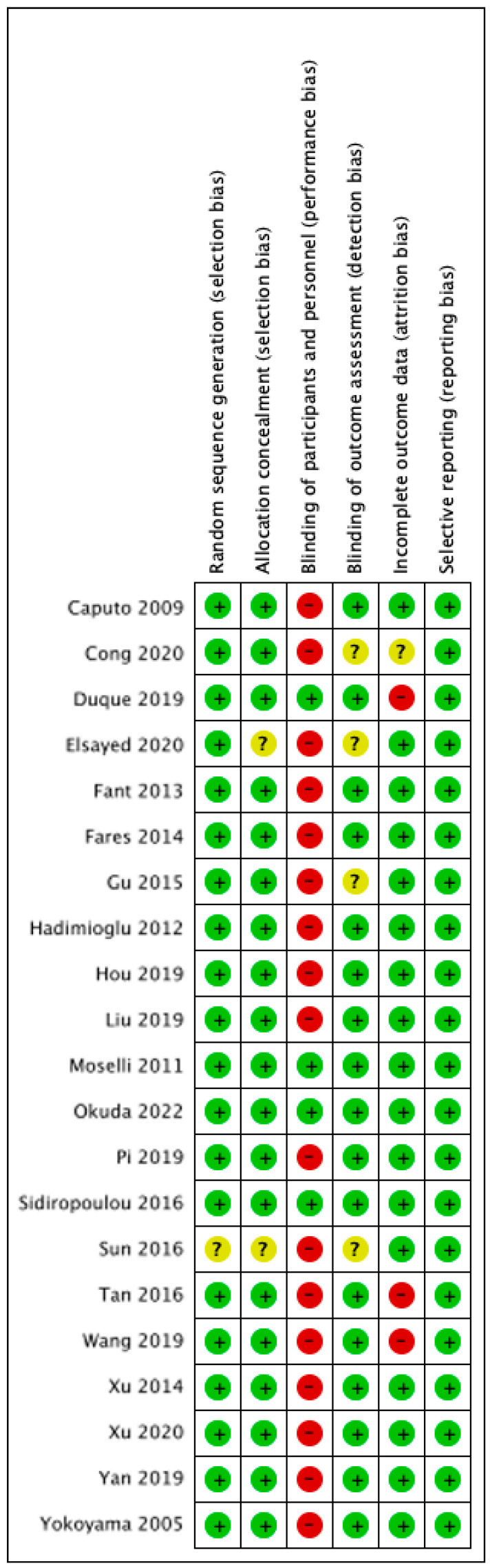
Risk of bias assessment. Green: low risk, amber: some concerns, red: high risk [22,23,24,25,26,27,28,29,30,31,32,33,34,35,36,37,38,39,40,41,42].

**Figure 3 cancers-17-01667-f003:**
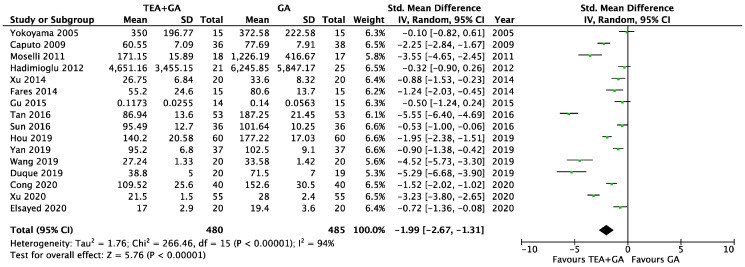
Forest plots of IL-6 between GA and GA plus EA cohorts [22,23,24,25,27,28,29,30,32,35,36,37,38,39,40,41].

**Figure 4 cancers-17-01667-f004:**
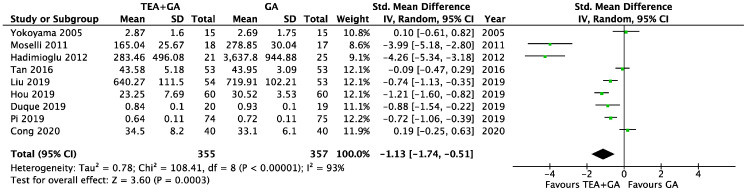
Forest plots of TNF-α between GA and GA plus EA cohorts [23,24,29,30,31,32,33,36,41].

**Figure 5 cancers-17-01667-f005:**
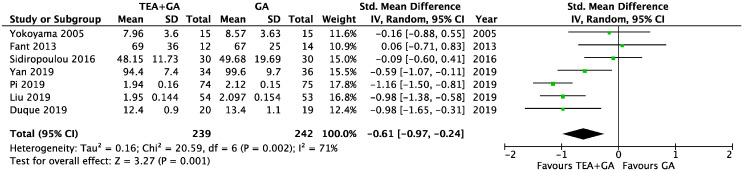
Forest plots of CRP between GA and GA plus EA cohorts [24,26,31,33,34,40,41].

**Table 1 cancers-17-01667-t001:** Characteristics of the included studies.

Study ID	Methods	Participants	Interventions	TEA Procedure	Surgery Duration (Min with SD or Range)		Outcomes
					TEA + GA	GA	
Caputo 2009 [22]	RCT, observer blinded	74 adults undergoing off-pump coronary artery bypass graft surgery	T 2/3 or T 3/4 epidural plus GA vs. GA alone	Perioperative, initial bolus of 5 mL bupivacaine 0.5% + 5 mL bolus followed by continuous infusion of 0.125% bupivacaine and 0.0003% clonidine at rate of 10 mL/h. Postoperative, top-up bolus doses 4 mL 0.25% bupivacaine for 72 h	248 (255–528)	361 (290–540)	troponin I, 8-isoprostane, cortisol, C3alpha, IL-6, IL-8, and IL-10
Cong 2020 [23]	RCT	120 adults undergoing radical resection of esophageal cancer	T 6/7 epidural + GA vs. PVB + GA vs. GA alone	Epidural catheter with 10 mL 0.375% ropivacaine	847 (306)	1102 (450)	IL-6, IL-4, TNF-α, IFN-γ, CD3+, CD4+, CD8+, CD4+/CD8+
Duque 2019 [24]	RCT, patients and observer blinded	39 adults scheduled for colorectal cancer surgery	T 7–10 epidural + GA vs. GA alone	Perioperative, initial dose of 0.5% bupivacaine 8–12 mL followed by continuous infusion of 0.5% bupivacaine at 7–10 mL/h. Postoperative, patient-controlled epidural infusions of 0.125% bupivacaine at 7–10 mL/h with additional boluses of 7 mL of same solution, with max of 2 boluses/h.	186 (58)	183.4 (79)	CPR, Cortisol, CCL2, TNF-α, IL-1, IL-2, IL-4, IL-6, IL-8, Nitric oxide, VEGF, MMP-3, Leucocytes, and Lymphocytes
Elsayed 2020 [25]	RCT	40 adults undergoing laparoscopic cholecystectomy	T 8/9 epidural + GA vs. GA alone	Epidural catheter with a bolus dose of 6–8 mL of 0.25% bupivacaine and 50 μg fentanyl. Continuous epidural infusion with 0.125% bupivacaine and fentanyl 1.5 μg/mL at 0.1 mL/kg/h.	68 (9.89)	74 (11.88)	IL-6, IL-8
Fant 2013 [26]	RCT, observer blinded	26 adults undergoing elective radical retropubic prostatectomy	T 10–12 epidural + GA vs. GA alone	Perioperative, epidural catheter with a bolus dose of 2–3 mL 0.5% bupivacaine with epinephrine followed by a continuous infusion of bupivacaine 0.5% at 2–4 mL/h	100 (27)	109 (30)	CRP, cortisol, IL-6, TNF-alpha, IFN-gamma, IL-2, IL-12p70, IL-17, IL-4, IL-10,
Fares 2014 [27]	RCT, observer blinded	30 adults scheduled for elective Ivor Lewis esophagectomy	TEA + GA vs. GA alone	Perioperative, initial dose of 0.1 mL/kg of 0.125% bupivacaine + fentanyl 15 μL/mL followed by continuous infusion of 0.1 mL/kg/h of 0.125% bupivacaine + fentanyl 10 μL/mL. Postoperative, continuous infusion of 0.1 mg/kg/h of 0.125% bupivacaine + fentanyl 5 μg/mL for 72 h.	270.7 (20.4)	269 (20.9)	IL-6, IL-8
Gu 2015 [28]	RCT	29 esophageal carcinoma patients undergoing thoracic surgery	T 7/8 epidural + GA vs. GA alone	Perioperative, continues infusion of 0.25% ropivacaine at 5–7 mL/h. Postoperative, continuous infusion of 0.125% ropivacaine + fentanyl 2 μL/mL at initial 10 mL followed by 5 mL.	167.6 (14.5)	164.8 (13.2)	cortisol, IL-6, IFN-γ, IL-4, IL-17
Hadimioglu 2012 [29]	RCT, observer blinded	46 adults scheduled for renal transplantation surgery	L1/2 epidural + GA vs. GA alone	Epidural: 0.5% bupivacaine (14–18 mL). Patient-controlled analgesia with morphine: 1 mg of loading dose, 0.5 mg/h basal infusion, 1 mg bolus doses with 20 min lock-out interval	102.1 (25.2)	100.9 (15.2)	TNF-α, IL-6, adiponectin, resistin
Hou 2019 [30]	RCT, observer blinded	120 adults undergoing colon cancer surgery	T 10/11 epidural + GA vs. GA alone	initial dose of 8 mL 0.375% bupivacaine followed by 4 mL bupivacaine/50 min			TNF-α, IL-6, IL-10, crotisol, adrenocorticotropic hormone, endothelin-1
Liu 2019 [31]	RCT, observer blinded	107 adults with early-stage gastric cancer undergoing tumor resection	T 8–10 epidural + GA vs. GA alone	Perioperative, epidural administration of 1% lidocaine (5–10 mL) and 0.375% ropivacaine using micropump (5–8 mL/h). Postoperative, injected 0.12% ropivacaine and 0.2 μg/mL sufentanil at rate of 5 mL/h, performed for 2–6 days.	205.02 (52.22)	211.72 (44.34)	CD4+ cell, CD8+ cell, IL-1, CRP, TNF-α, IL-8
Moselli 2011 [32]	RCT, observer blinded	35 adults undergoing major surgery for colon cancer	T 9/10 or T 12/L1 epidural + GA vs. GA alone	Initial dose of 5–7 μg of sufentanil in fractioned boli and 0.5% L-bupivacaine 7–9 mL/h in continuous infusion started 40 min before skin incision followed by continuous epidural infusion of 0.5% L-bupivacaine 3–5 mL/h.	243.8 (88.3)	216.5 (105.3)	IL-4, IL-10, IL-6
Pi 2019 [33]	RCT, observer blinded	149 adults with early-stage non-small cell lung carcinoma undergoing tumor resection	T 8–T 10 epidural + GA vs. GA alone	Epidural administration of 1% lidocaine (5–10 mL) and 0.375% ropivacaine using a micropump at rate of 5–8 mL/h.	219.02 (45.46)	217.72 (36.65)	IL-1, IL-8, CRP, TNF-α
Sidiropoulou 2016 [34]	RCT, patients and observer blinded	60 adults scheduled to undergo elective laparoscopic cholecystectomy	T 12–L2 epidural + GA vs. GA alone	Epidural catheter with a bolus dose of ropivacaine 1% 15 mL	64.1 (11.8)	65.2 (16.3)	CRP, cortisol
Sun 2016 [35]	RCT, observer blinded	72 adults with hepatocellular carcinoma undergoing hepatectomy	T 8/9 epidural + GA vs. GA alone	Epidural with 1% lidocaine 5 mL/h	152.81 (13.37)	154.86 (13.63)	cortisol, CD3+, CD4+ T cell, IL-6, IL-10, IFN-γ
Tan 2016 [36]	RCT, observer blinded	106 adults undergoing lobectomy or pneumonectomy for lung cancer	T 5–7 epidural + GA vs. GA + dexmedetomidine vs. GA alone	Initial dose of 8–10 mL 0.25% ropivacaine followed by 0.15% ropivacaine and 1.79 μg/mL fentanyl at 5 mL/h and bolus dose of 6 mL without 40 min lockout time.			IL-6, TNF-α
Wang 2019 [37]	RCT, observer blinded	40 adults undergoing radical resection of gastric cancer	T 8–T 10 epidural + GA vs. GA alone	Initial dose 0.5% ropivacaine, 5–7 mL followed by 0.2% ropivacaine and 0.5 microg/mL sufentanil was available for 72–120 h	135 (120, 210)	135 (120, 225)	CD3+ cell, CD4+, CD8+, IL-4, IL-6, IFN-γ
Xu 2014 [38]	RCT, observer blinded	40 adults undergoing open colon cancer surgery	T 9–T 12 epidural + GA vs. GA alone	Perioperative, initial dose of 0.375% ropivacaine was 6–8 mL followed by continues infusion of ropivacaine at 5 mL/h. Postoperative, continuous infusion of 4 mL/h 0.15% ropivacaine and 0.5 μg/mL sufentanil and 2 mL bolus on request	102 (29)	118 (38)	VEGF-C, TGF-β, IL-6, IL-10
Xu 2020 [39]	RCT, observer blinded	110 adults undergoing laparoscopic colorectal cancer surgery	T 9–T 12 epidural + GA vs. GA alone vs. GA + TAP block	Perioperative, initial dose of 6–8 mL 0.25% ropivacaine followed by continuous infusion rate of 5 mL/h. Postoperative, 0.15% ropivacaine and 0.5 μg/mL sufentanil at continuous infusion rate of 4 mL/h	132.2 (19.6)	131.0 (20.2)	VEGF-C, IL-6, adrenaline, cortisol
Yan 2019 [40]	RCT, observer blinded	74 adults undergoing radical resection of cervical carcinoma	L2/3 epidural + GA vs. GA alone	Perioperative, infused 10 mL 0.75% ropivacaine. Postoperative, 0.125% ropivacaine; background dose, 8 mL/h; PCA dose, 2 mL/time, at 20 min intervals	145.6 (13.5)	151.3 (14.8)	CRP, IL-6, neutrophils
Yokoyama 2005 [41]	RCT, observer blinded	30 adults undergoing radical esophagectomy	T 3/4 plus T 10/11 epidural + GA vs. GA alone	Perioperative, epidural catheters with 8 mL 1.5% lidocaine. Postoperative, 1% lidocaine with 4 μg/mL fentanyl at 6 mL/h	616 (69)	648 (66)	epinephrine, norepinephrine, cortisol, ACTH, IL-1, IL-6, IL-10, TNF-α, B cells, T cells, NK cells, CRP
Okuda 2022 [42]	RCT, patients and observer blinded	60 adults undergo lung cancer surgery	Th5/Th8 epidural + GA vs. GA alone	Perioperative, initial dose of 5–15 mL 0.25% levobupivacaine followed by continuous infusion at the rate of 3–10 mL/h. Postoperative, 5–10 mL 0.25% levobupivacaine with patient-controlled epidural analgesia in both groups	121 (40)	121 (38)	IL-6, TNF-α, IL-10, MDA

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
