# Peer review of "Combined General/Epidural Anesthesia vs. General Anesthesia on Postoperative Cytokines: A Review and Meta-Analysis"

_cancers, 2025, doi:10.3390/cancers17101667_

Round 1
Reviewer 1 Report
Comments and Suggestions for Authors
Review on Combined GEA vs. GA on postoperative cytokines: a systematic review and meta-analysis.
The study seems well conducted and is clearly described, with the provided supplemental material strengthening the main part of the manuscript.
Major concerns:
It is doubtful whether laparoscopic cholecystectomy has the same effects on the human body (e.g. immune response) as off-pump CABG. Furthermore, the renal transplant patients included are hopefully the donors and not the acceptors, as the last will be treated with drugs influencing the immune system to prohibit immediate rejection of the organ. (If correct, make changes accordingly in Table 1).
Nothing in this study is stated on the drugs used via the epidural in the included studies. Although this wasn’t a part of the search strategy, for proper interpretation it is necessary to have some idea whether only local anaesthetics were used, or local anaesthetics were combined with opioids, or perhaps even opioids as mono-drug has been used in the epidurals. This item should at least be addressed in the discussion section.
No information is given on the timeperiod proper epidural analgesia was present in the postoperative phase. These items (both, timeperiod as well as delivering proper analgesia by EA) should also be addressed in the discussion section.
Minor concerns:
At line 50: risk of in infectious complications, should be, risk of infectious complications.
At line 141, for the first time it is described to the reader that both cancer and non-cancer studies have been implanted for the analysis. It is better to add this information for the reader in the abstract, as well as in line 89.
Line 338: are increasing evidence, should be, is increasing evidence.
Line 338 event, should be, events.
Line 338 and 342: affect, seems to be meant effect.
Reviewer 2 Report
Comments and Suggestions for Authors
Title: Combined General/Epidural Anesthesia vs General Anesthesia on Postoperative Cytokines: A Systematic Review and Meta-Analysis
The present study aimed to analyze inflammatory cytokines levels after surgery under combined epidural/general anesthesia (EA+GA) vs general anesthesia (GA). Comments related to the review article are mentioned below.
- The results of this systematic review and meta-analysis focus on 24 hours after surgery; the authors should include the serum half-lives of all analytes, including cytokines, with recent supporting references. This should be added to the introduction section.
- Apart from the cancer surgeries, were any studies related to another surgical process also included for analysis? If yes, elaborate, giving the reasons for the specific selection.
- Authors mention, “Finally, we included 21 studies in our meta-analysis, and their characteristics are described in Table 1 [22-42]. Among them, 15 are cancer surgeries and seven are non-cancer surgeries.” Elaborate on the types of these non-cancer surgeries in the methodology section.
- Also, mention the reason for the type of cancers included in this study.
- Add what would be the effect on the markers analysed here based on the type of surgery, duration, gender, and age of the patient. Is it only the type of anaesthesia given that influences the serum levels of the analytes mentioned in this study?
- What was the correlation between the dose of anaesthesia and the proinflammatory cytokines? This needs to be addressed, as the dose may affect serum cytokine levels.
- Though the authors have provided an in-depth analysis, what could be the translation value for the findings of this piece of research? Will the serum profiles obtained for ED+GA and only GA in any case be considered a deciding factor for the choice of anaesthesia?
- The study needs to elaborate on the biases induced due to the selection process.
Round 2
Reviewer 1 Report
Comments and Suggestions for Authors
The study is well described and OK in the present form
Reviewer 2 Report
Comments and Suggestions for Authors
The authors satisfactorily answered my queries and incorporated the suggested changes. The revised version of the manuscript may be accepted for publication.